# A deep learning-based hybrid model of global terrestrial evaporation

Akash Koppa [1✉], Dominik Rains[1], Petra Hulsman [1], Rafael Poyatos [2,3] & Diego G. Miralles [1]

Terrestrial evaporation ($E$) is a key climatic variable that is controlled by a plethora of environmental factors. The constraints that modulate the evaporation from plant leaves (or transpiration, $E_t$) are particularly complex, yet are often assumed to interact linearly in global models due to our limited knowledge based on local studies. Here, we train deep learning algorithms using eddy covariance and sap flow data together with satellite observations, aiming to model transpiration stress ($S_t$), i.e., the reduction of $E_t$ from its theoretical maximum. Then, we embed the new $S_t$ formulation within a process-based model of $E$ to yield a global hybrid $E$ model. In this hybrid model, the $S_t$ formulation is bidirectionally coupled to the host model at daily timescales. Comparisons against in situ data and satellite-based proxies demonstrate an enhanced ability to estimate $S_t$ and $E$ globally. The proposed framework may be extended to improve the estimation of $E$ in Earth System Models and enhance our understanding of this crucial climatic variable.

[1] Hydro-Climate Extremes Lab (H-CEL), Ghent University, Ghent, Belgium. [2] CREAF, Catalonia, Spain. [3] Universitat Autònoma de Barcelona, Catalonia, Spain.
✉email: akash.koppa@ugent.be

$E$ is a key element of the global water cycle: approximately two-thirds of the precipitation over land is evaporated back into the atmosphere[1]. Due to its influence on water vapor and cloud feedbacks, $E$ plays a crucial role in global warming, and its projected increase is expected to intensify the global hydrological cycle[2]. Changes in $E$ will not only have far-reaching consequences on water availability and climate[3,4], but can also severely affect the occurrence of hydroclimatic extremes[5] and the ability of ecosystems and river basins to recover from them[6–8]. Moreover, $E$ is an important indicator of vegetation stress, thus it is widely used for estimating drought conditions[9] and their implications for water management, ecosystem health, and agricultural production[10]. Its reliable representation in hydrological and climate models is therefore crucial, and so is its accurate global monitoring from space. However, $E$ cannot be derived directly from satellite-based measurements, which is why even retrieval algorithms tend to rely on process-based formulations[11].

Several approaches exist to estimate $E$ at large scales using process-based models. Some of them simulate $E$ as a residual of the energy balance, or derive it empirically using vegetation, temperature, and radiation data. These approaches are primarily employed in high-resolution remote sensing, especially in agricultural areas, owing to its minimal input data requirements[12,13]. Other models employ a flux-based approach to derive $E$ using more physically-founded methods, such as the Monin-Obukhov similarity theory, to calculate the gradients of specific humidity between the atmosphere and land surface (vegetated or non-vegetated), and explicitly model the surface resistance to the diffusion of water vapor. This approach is prevalent in climate models[14]. Finally, the third approach, commonly used in hydrological models[15] and satellite retrieval algorithms[16–18], involves the prior calculation of potential evaporation ($E_p$), a theoretical maximum for the given land cover and meteorological conditions. Subsequently, actual $E$ is estimated by reducing $E_p$ by a certain factor that accounts for the sub-optimal conditions of evaporation due to (e.g.,) water scarcity; this is referred to here as 'evaporative stress' ($S$), and more precisely 'transpiration stress' ($S_t$) when applied to plant transpiration. Independent of the approach, significant uncertainties remain in the current global estimates of $E$, and this applies to both climate models[19] and satellite-based algorithms[20].

In this study, we focus on stress-based models of $E$, the most common approach to derive global $E$ from satellite data[21]. In such models, uncertainty arises from the formulations of $E_p$ and $S$ (and particularly $S_t$). While several process-based formulations of $E_p$ exist[22,23], they differ in their estimates substantially, and even the mere definition of $E_p$ as a concept remains elusive[24]. Nevertheless, the chosen $E_p$ function forms the most process-based part of the stress-based $E$ models, and while parameters within $E_p$ formulations can be better constrained with more data[25], the opportunities to improve stress-based models via modifications to $E_p$ remain limited[26]. Therefore, we focus on the main source of uncertainty: the $S$ formulation. This uncertainty arises from the lack of understanding of the response of plant transpiration (the major source of $E$ in vegetated ecosystems) to environmental stressors, particularly at the spatial resolution at which global models operate. Here, we note that the focus of this study is the stress which limits vegetation transpiration below the atmospheric potential, and therefore can be triggered even under conditions in which plants do not experience stress from a physiological standpoint. The transpiration stress (i.e., $S_t$) should encapsulate multiple interacting hydroclimatic variables that affect different aspects of plant physiology and structure which affect transpiration in a highly non-linear manner at multiple time scales[27]. However, $S_t$ formulations used in existing global models are simple, not capturing all the influences and interactions among the stressors. This occurs because they are based on a limited number of experimental studies whose extrapolation to global scale is hindered by their local nature[28–30]. The complexity of the interactions among these stressors, and the fact that they involve physiological processes that are unobserved, calls for machine learning techniques as a suitable solution to this long-standing challenge.

Machine learning methods have become popular in Earth sciences in recent years, enabling the discrete classification of important geo-spatial variables which are hard to map, such as clouds[31], soils[32], and forest cover[33]; but also estimation of dynamic variables, such as carbon fluxes[34], precipitation[35], or river discharge[36]. In fact, machine learning models trained on in situ measurements of $E$ and other hydro-meteorological covariates, have already been used to estimate global $E$[34]. However, pure machine learning–based approaches have several disadvantages in realistically modeling Earth system processes. Machine learning models do not obey the physical limits which constrain different scales, such as the closure of water and energy balances. Further, the black-box nature of machine learning hinders the interpretability, an important requirement if the influence of individual covariates needs to be realistically represented to improve process understanding. More importantly, the use of pure machine learning methods for specifically estimating $E_t$ at global scales is hindered by the fact that in situ observations of $E_t$ have a small footprint which is not representative of $E_t$ at the coarser grid scales at which global models operate.

An emerging research direction, and the approach adopted in this study, is to combine process-based and machine learning models in a symbiotic manner. 'Hybrid' models retain the advantages of process-based models, i.e., physical consistency and interpretability, and those of machine learning models, i.e., more realistic data-driven formulations of processes that are insufficiently understood[37]. Several proof-of-concept implementations have demonstrated the advantages of hybrid modeling in climate sciences with machine learning sub-models employed for representing different processes[38,39] or for improved model parameter discovery[40]. For modeling $E$ in particular, attempts have been made to physically constrain pure machine learning models to improve the accuracy of $E$ estimates[25,41]. However, an important research question is whether hybrid models are capable of operating at a global scale with machine learning used to replace specific process formulations.

Here machine learning, specifically deep learning, is used to learn the functional relationship between covariates ($S_t$ drivers) and target process ($S_t$). We exploit recent progress in satellite-based remote sensing and an unprecedented number of in situ observations spread across the globe to develop a novel formulation of $S_t$ from the ground-up without any prior assumptions. Therefore, the objective of using deep learning is fundamentally different (development of an improved formulation of the transpiration stress response of vegetation) from that of purely machine learning-based models which are designed to predict $E_t$ directly and suffer from scaling issues described above. Further, we implement the new formulation of $S_t$, and execute it online, in a process-based model of global evaporation which provides physical constraints to the deep learning-based $S_t$ formulation. In doing so, we develop a hybrid model in which the new deep learning-based formulations of $S_t$ are tightly coupled to the process-based, aiming to simulate daily $E$ at the global scale. A comprehensive evaluation of the model is carried out using in situ observations and gridded datasets, including comparisons to pure machine learning and process-based approaches.

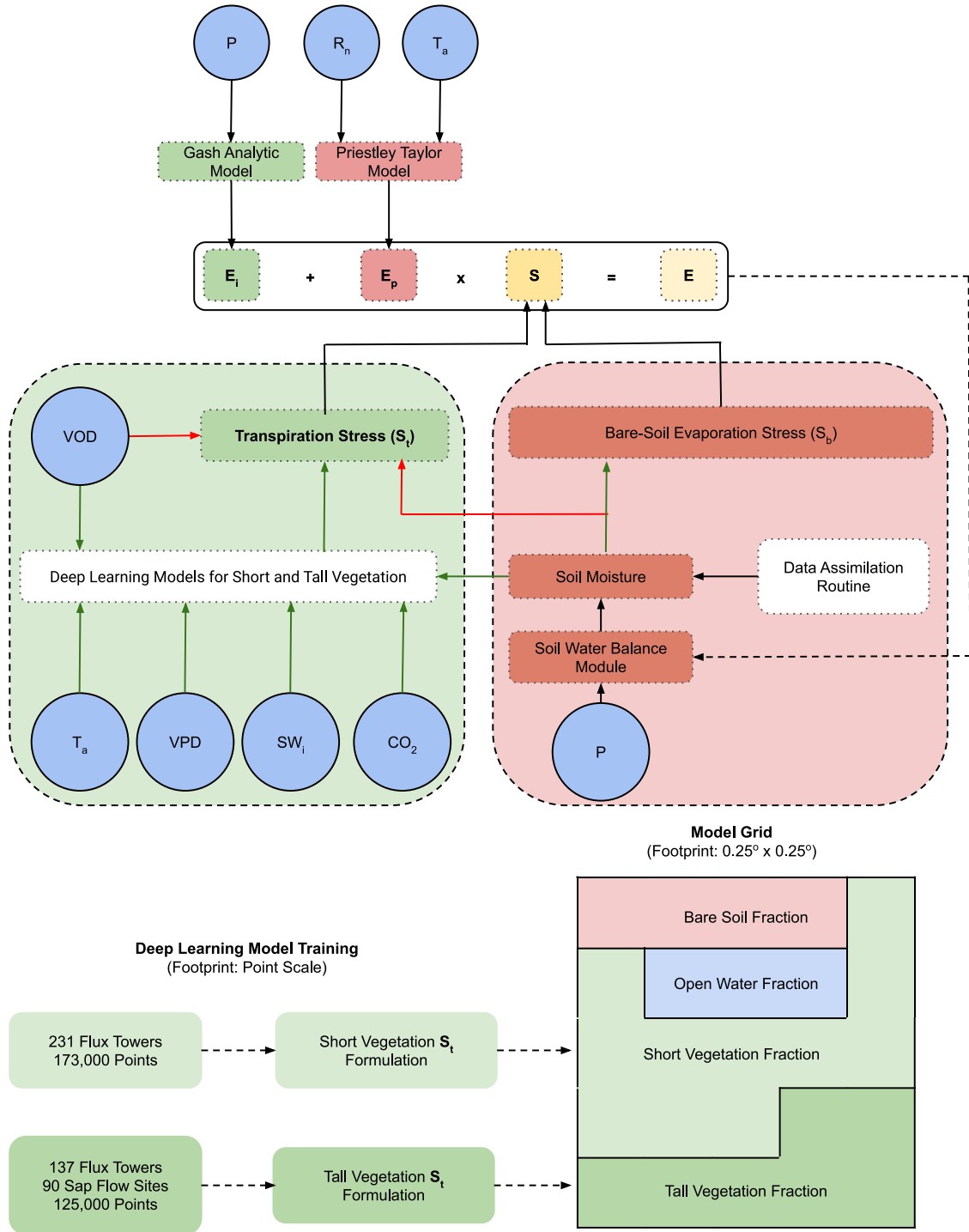

**Fig. 1 Hybrid model architecture.** Schematic of the hybrid terrestrial evaporation model, including the representation sub-grid heterogeneity and the difference in the footprints of the deep learning model and the hybrid model. $E_i$ is interception, $E_p$ is potential evaporation, $S$ is the evaporative stress factor, $S_t$ is transpiration stress, $E$ is actual evaporation, $P$ is precipitation, $R_n$ is net radiation, $T_a$ is air temperature, $VOD$ is vegetation optical depth, $VPD$ is vapor pressure deficit, $SW_i$ is incoming shortwave radiation, and $CO_2$ is carbon dioxide. The red arrows indicate modeling steps which are exclusive to the processed-based model, the green arrows are steps which have been added in the hybrid, and the black arrows are steps common to both the models.

## Results

**Hybrid model architecture.** A hybrid model at the highest level of abstraction is made up of two components: a process-based host model and machine learning-based formulations embedded in the host for representing certain processes[37]. For the process-based model, we choose the Global Land Evaporation Amsterdam model (GLEAM)[16,42]. GLEAM simulates $E$ as a summation of its

constituents: $E_t$, bare-soil evaporation ($E_b$), open water evaporation ($E_w$), snow sublimation ($E_s$), and interception loss ($E_i$). $E_t$ and $E_b$ are estimated for every grid cell of the global model using a Priestley Taylor-based formulation for $E_p$ and their respective evaporative stress factors ($S_t$ and $S_b$), weighted by the fractional coverage of short vegetation, tall vegetation, bare-soil, and open water (Fig. 1). Interception is based on the Gash analytical

model[43]. GLEAM contains a multi-layer soil water balance model in which satellite-based surface soil moisture data are assimilated. $S_b$ is a function of soil moisture content (see Methods), while $S_t$ accounts for the transpiration stress caused by shortage of plant available water ($PAW$) and sub-optimal phenological state (represented by vegetation optical depth, $VOD$). In nature, however, several additional stressors are responsible for limiting $E_t$ below its potential, which are not considered within the $S_t$ formulation of the process-based GLEAM. The exact responses of $E_t$ to these stressors are ecosystem-dependent and difficult to encapsulate into a single stress factor ($S_t$).

Here, using deep learning and reliable field observations, we aim to recover an $S_t$ that correctly encodes the functional relationships among the multiple stressors existing in nature. Deep learning models are developed at daily time scales using observational data from a large network of eddy covariance or flux towers and sap flow measurements. The models are developed separately for short (231 flux towers and 173,000 data points) and tall vegetation (137 flux towers and 90 sap flow measurement sites, 125,000 data points) (see Methods for the details of the target variable and covariates used in the deep learning models). We consider four other transpiration stressors, in addition to $PAW$ and $VOD$, that are known to regulate stomatal conductance and hence influence $S_t$: (a) vapor pressure deficit ($VPD$), as an indicator of atmospheric dryness[44], (b) air temperature ($T_a$), to include the effects of sub-optimal temperature and heat stress[45], (c) incoming shortwave radiation ($SW_i$), to incorporate the influence of light limitation[46], and (d) atmospheric carbon dioxide ($CO_2$) concentration, which exhibits a first order control on stomatal opening[47]. We note that the slowly evolving effects on transpiration of long-term ecological or plant trait adaptation in response to rising $CO_2$ (as reflected on water use efficiency trends) may not be adequately captured by training the machine learning algorithms on the limited record length of available flux tower and sap flow measurements[48]. The potential effect of phosphorous and nitrogen limitations on $S_t$[49] is not considered in this study due to the lack of dynamic global data. In addition, the influence of plant traits such as root depth, isohydricity, and other anatomical and morphological traits, and their fine-scale or inter-species variations is not explicitly considered, since reliable data for upscaling such traits so that they can be implemented within a global model is not available.

Finally, the hybrid model of global $E$ is created by coupling the deep learning-based model of $S_t$ to the GLEAM process-based model. At every (daily) time step, and at every (0.25 degree) grid cell of the global model, the soil water balance module of GLEAM uses precipitation ($P$) to compute $PAW$. Then, $PAW$, $VOD$, $T_a$, $VPD$, $SW_i$, $CO_2$ are transferred to the (offline-trained) deep learning model (see Methods). The deep learning model is run in predictive mode to generate $S_t$. $S_t$ is then used to constrain $E_p$ and thus compute $E$ by the process-based host model. Finally, $E$ is used to update the soil moisture (and $PAW$) before the process is repeated for next time step (Fig. 1).

**Validation with in situ measurements**. $S_t$ and $E$ estimates from the hybrid model are validated at 458 in situ monitoring stations (see Figs. 12 and 13 in Supplementary Information) sourced from several flux tower and sap flow databases (refer to the Methods section for the calculation of $S_t$ from flux tower and sap flow data). The hybrid model performance is compared to that of the fully process-based model. Violin plots and spatial maps illustrate the Kling-Gupta Efficiency (KGE), a metric which combines correlation, variability bias, and mean bias (see Methods). KGE values theoretically range from $-\infty$ to 1.0, with values greater -0.41 implying that the model is a better predictor than the mean seasonal cycle[50].

The violin plots (Fig. 2a) show the distribution of KGE values calculated for the 231 stations located in short vegetation ecosystems, and the 227 stations in located in tall vegetation ecosystems (137 flux tower sites and 90 sap flow measurement stations). We see that both the process-based model and the hybrid model accurately estimate $S_t$ in short vegetation ecosystems (including Croplands, Shrub and Grasslands, and Wetlands) and tall vegetation ecosystems (consisting of Broadleaf, Needleleaf, and Mixed forests)—see Table 3 in Supplementary Information for station-wise land cover classification. For most stations (>75%), KGE values from the process-based model are higher than $-0.41$. However, the deep learning model of $S_t$ improves these results, particularly over tall vegetation—see Fig. 2a. The higher KGE is attributable to improvements in the bias and variability components of the KGE rather than the correlation component—refer to Figure 1 in Supplementary Information for violin plots of correlation and root mean square error (RMSE). While the average correlations of the process-based model estimates of $S_t$ are similar to those by the hybrid model, the RMSE of the hybrid model tends to be substantially lower, particularly for tall vegetation ecosystems.

Next, we check whether the improvement in the estimation of $S_t$ in the hybrid model is propagated to the simulation of $E$. From Fig. 2b, it is evident that the improvements in $S_t$ are not linearly translated to $E$. This can be attributed to the fact that the vast majority of the flux towers and sap flow sites are in energy-limited regions, where $E$ dynamics are influenced more by $E_p$ than by $S_t$. Overall, both models exhibit high, and similar, KGE values (median value of approximately 0.5) for short vegetation. For tall vegetation, the hybrid model outperforms the process-based model in terms of KGE values. In terms of correlation and RMSE, both models perform similarly (see Fig. 1 in Supplementary Information): the process-based model exhibits marginally higher correlations, while the RMSE of the hybrid model is lower for both vegetation classes. We also compare the estimates of $E$ from the hybrid model with that of a purely machine learning-based dataset, FLUXCOM (Fig. 2 in Supplementary Information). We see that while the overall performance of both approaches is similar, the hybrid model tends to outperform FLUXCOM in forest (tall vegetation) ecosystems.

To understand the difference between the hybrid and process-based models better, we compare the spatial distribution of differences in KGE values for $S_t$ and $E$ estimates from the two models for different geographical zones (Fig. 3, also see Figs. 3 and 4 in Supplementary Information for absolute values of KGE for $S_t$ and $E$). In North America (NA), which has the largest number of flux towers and sap flow sites, the hybrid model outperforms the process-based model in estimating $S_t$ and $E$, especially in the humid eastern and north-eastern areas. In comparison, both models tend to inaccurately simulate $S_t$ in the arid south-west region. In Europe (EU), the hybrid model performs better than the process-based model in estimating $S_t$ across the majority of the flux tower stations, including stations which are located in the relatively arid south. However, in Asia (AS) and rest of the world (RW), the performance of the hybrid model is very similar to the process-based model. One reason could be that the AS and RW regions have a very sparse distribution, and thus flux towers and sap flow sites in those ecosystems may have distinct biophysical characteristics from the majority of sites in the training database. Further, we compare the spatial maps of correlation and RMSE (see Figs. 5–8 in Supplementary Information) to understand the source of the disparity in KGE values. In terms of correlation, the two models perform very similarly to each other across the different regions. Therefore, the major source of improvement in the hybrid model can be traced to the better estimation of the variability seen in the

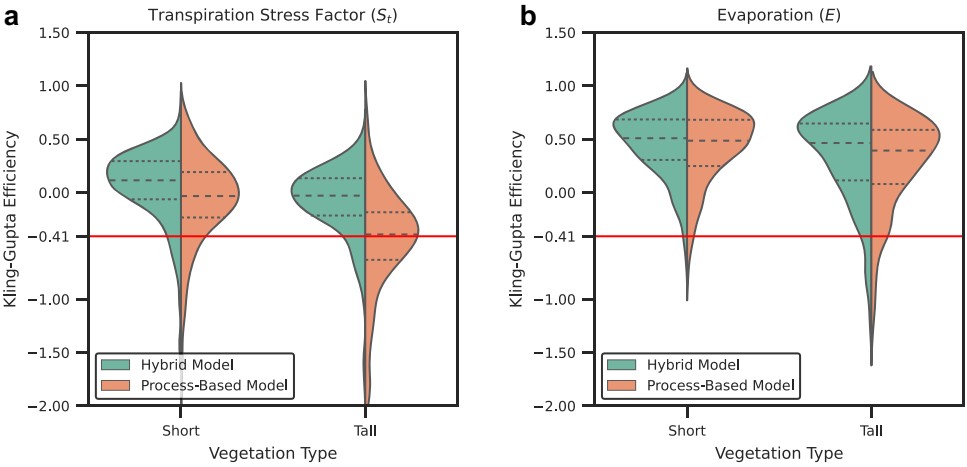

**Fig. 2 Summary of in situ validation of the hybrid and process-based models. a**, **b** Violin plots showing the distribution of the Kling-Gupta Efficiency (KGE) metric for the transpiration stress factor ($S_t$) and evaporation ($E$), respectively, calculated for all flux tower and sap flow measurement sites. The KGE distribution for the hybrid and process-based models are classified according to short and tall vegetation types. The dashed lines represent the median (large dashes) and the interquartile range (small dashes). The red line represents a KGE value of -0.41, above which a model prediction or simulation is considered better than the mean seasonal cycle. For the sap flow sites, transpiration estimates ($E_t$) instead of $E$ are used.

**Fig. 3 In situ comparison of the hybrid and process-based models.** Maps showing the difference in the Kling-Gupta Efficiency (KGE) metric between the hybrid model and process-based model for the transpiration stress factor ($S_t$) and evaporation ($E$) calculated using observations at flux tower and sap flow measurement sites in different geographical zones: North America (NA), Asia (AS), Europe (EU), Rest of the World (RW). Blue (red) tones indicate an improvement (degradation) in the hybrid model compared to the process-based counterpart. For the sap flow sites, transpiration estimates ($E_t$) instead of $E$ are used.

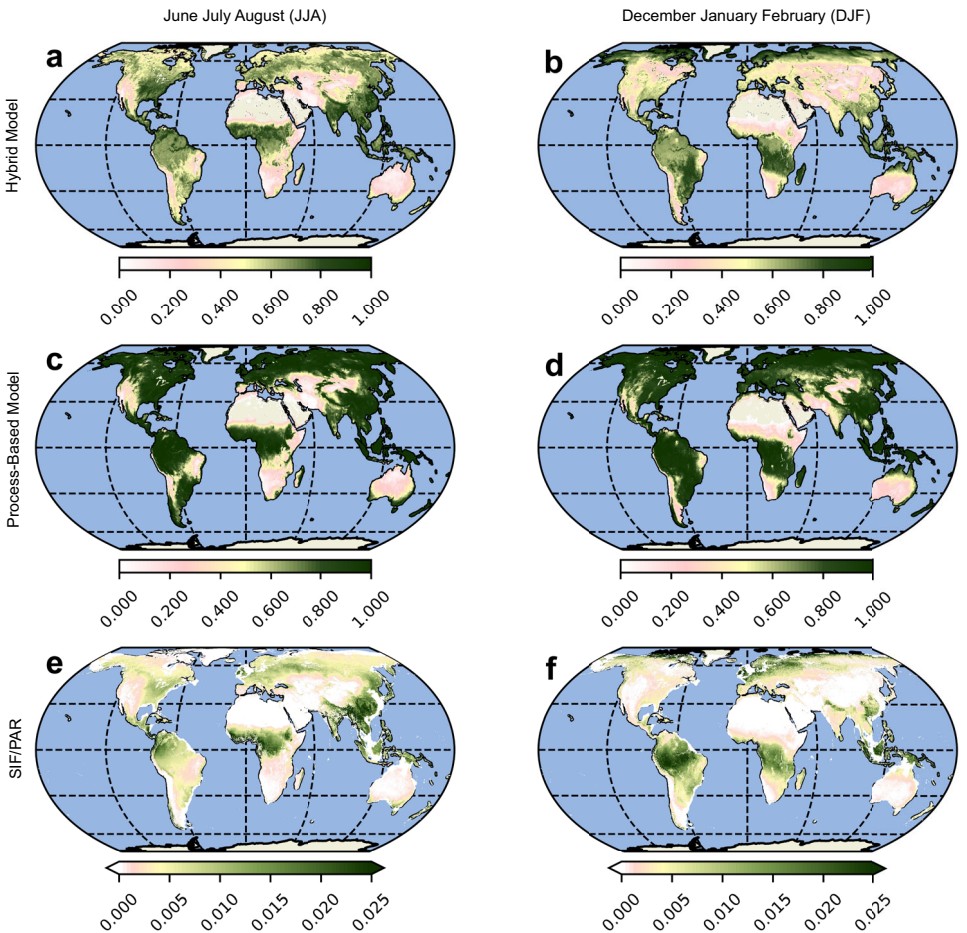

**Fig. 4 Global scale evaluation of modeled transpiration stress.** Comparison of the seasonal mean transpiration stress factor ($S_t$) from the processed-based and hybrid models and the ratio of solar-induced chlorophyll fluorescence and photosynthetically-active radiation (*SIF/PAR*) for June-July-August (JJA) **a**, **c**, **e** and December-January-February (DJF) **b**, **d**, **f** seasons. Note: The unit of measurement of *SIF* is $mWm^2/sr/nm$ whereas *PAR* is in $W/m^2$.

observation, a fact supported by the violin plots (Figure 1 in Supplementary Information). Further, we notice that the discrepancy in the $S_t$ estimates between the two models, does not translate to an improved $E$ estimation, particularly in energy-limited regions (Fig. 3), which are poorly represented in the training data. Finally, we also compare the performance of the hybrid model against FLUXCOM at individual flux tower and sap flow measurement sites (Fig. 9 in Supplementary Information). Similar to the comparison with the process-based model, we see that the hybrid model underperforms in the relatively arid western parts of the US and the Iberian Peninsula.

**Comparison with global datasets.** The goal of the hybrid model is to generate spatially and temporally continuous estimates of $S_t$ and $E$ over the entire continental surface. Therefore, it is important to also validate it against independent global estimates of both $S_t$ and $E$. Therefore, $S_t$ and $E_t$ seasonal aggregates are compared with other global datasets in Fig. 4 and Fig. 5, respectively. To further investigate the realism of these global patterns, the temporal dynamics are investigated in Fig. 6 by displaying correlation maps based on monthly time series.

Due to the absence of observations of $S_t$ at those scales, we choose a satellite-retrieved proxy that has been shown to represent the transpiration stress experienced by vegetation reasonably well: the ratio of solar-induced chlorophyll fluorescence to photosynthetically-active radiation (*SIF/PAR*)[51] (see Methods). We note here that the units and range of *SIF/PAR* values are

different from those of $S_t$, but that the spatial gradients and temporal dynamics are expected to be comparable. We also caution that the comparison may not be appropriate under extreme conditions and higher $CO_2$, where carbon and water cycles may decouple[52]. In June-July-August (JJA), summer season in the Northern Hemisphere, we see that the spatial patterns of $S_t$ in the hybrid model are similar to those in the process-based model (Fig. 4a, c). However, the hybrid model captures better the higher transpiration stress that is suggested by the low values of *SIF/PAR* in the higher latitudes (Fig. 4e). For December-January-February (DJF), the picture is similar; $S_t$ in the higher latitudes is accurately captured by the hybrid model (Fig. 4b,d,f). Similarly, we see that the hybrid model represents the stresses in the Congo, Amazonian and Eastern Asian rainforests accurately, both in JJA (Fig. 4a) as well as DJF (Fig. 4a). Figure 6a, c shows the temporal correspondence between $S_t$ and *SIF/PAR* for the hybrid and process-based models, respectively, while Fig. 6e shows the difference between the two previous maps. We see that the hybrid model exhibits a positive correlation with *SIF/PAR* over a majority of the continental surface with parts of Amazonia, Congo, and South East Asia (Fig. 6a) being an exception. The hybrid model $S_t$ shows a better correlation with *SIF/PAR* in eastern China and in northern latitudes—compare Fig. 6a, c. In contrast, the process-based model shows a higher correlation in large parts of western North America, Europe, and Australia. In addition, the hybrid model shows a marked improvement in the spatial correlation with *SIF/PAR*, both in the JJA season (0.66 vs. 0.59) and in the DJF season (0.42 vs. 0.34).

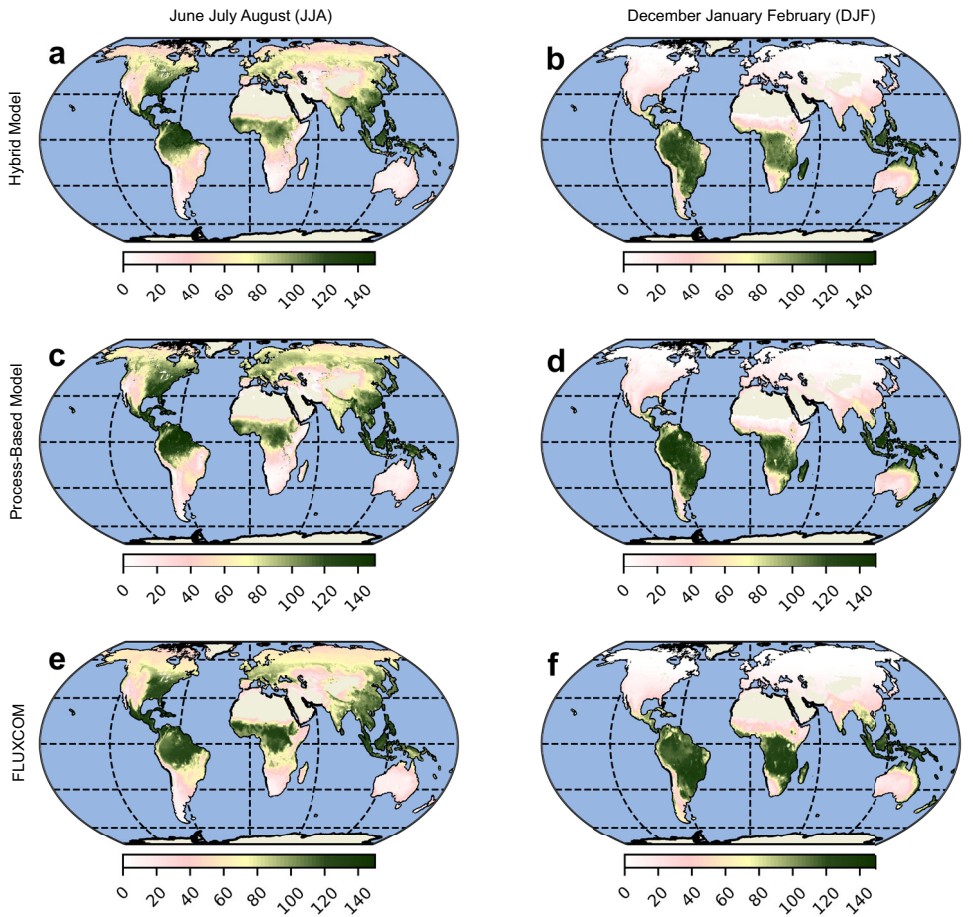

**Fig. 5 Global scale evaluation of modeled evaporation.** Comparison of the seasonal aggregates of evaporation (*E*) from the processed-based and hybrid models compared with a purely machine learning-based model trained directly on evaporation from FLUXNET sites as the target variable (FLUXCOM[34]) for JJA **a**, **c**, **e** and DJF **b**, **d**, **f** seasons. Note: The units of *E* is *mm/month*.

We also compare the *E* estimates from the hybrid and process-based models with a purely machine learning-based *E* dataset (FLUXCOM) which is trained on a subset of the global flux towers used in this study[34]. We see that in both seasons, JJA and DJF, the spatial patterns of *E* from the hybrid and process-based models are similar to those from FLUXCOM (Fig. 5). Regions of divergence are seen in the northeastern parts of South America, and southern and eastern Africa, where the FLUXCOM *E* estimates are higher than those of the hybrid and process-based models, especially during JJA. The correlation maps (Fig. 6b, d) show a high correspondence between the hybrid model estimates of *E* and FLUXCOM. A major region of divergence that stands out in both the hybrid and process-based models is Amazonia. This may relate to the fact that very few stations are available in tropical forests for model training, and therefore both the estimates of FLUXCOM and the hybrid model tend to be more uncertain there, and it may also reflect the lack of explicit consideration of interception loss as a component of *E* in FLUXCOM. Meanwhile, the difference between the correlations of the hybrid and process-based model with FLUXCOM is nominal (Fig. 6f). The hybrid model also shows mild improvements in the spatial correlation to FLUXCOM, both during JJA (0.84 vs. 0.81) and DJF (0.95 vs. 0.94).

## Discussion

The growing complexity of large-scale Earth system and climate models requires increasingly high computational resources. More importantly, processes are frequently represented based on limited experimental understanding and are thus uncertain in their application at larger scales. Hybrid modeling approaches have the potential to reduce the ill-effects of over-parameterization, reduce computation times, and even improve accuracy in process representation[53]. Here, we focus on one of the main unknowns in the global water cycle and a key variable in climate models: terrestrial evaporation (*E*). We developed and applied a global-scale hybrid model of *E*, in which a deep learning-based formulation of transpiration stress was embedded within a process-based model at daily timescales. We showed that the deep learning model, designed without a priori assumptions, and based on expert knowledge, is overall more accurate than the traditional process-based counterpart at capturing the non-linearly interacting processes that yield transpiration stress. The biggest improvement is seen in forested (tall vegetation) regions, especially in northern latitudes. This has important implications for constraining transpiration estimates in tropical, temperate, and boreal forests which contribute a major part of the global transpiration[54]. The study also highlights a limitation of any deep learning model, in which sufficient availability of training data is crucial: the majority of the flux towers and sap flow measurement sites used for training are located in North America and Europe. This is especially relevant for modeling Earth system processes that exhibit large regional (and local) variability, and thus for which the ability of any data-driven formulation to generalize over the entire globe will by default be imperfect. From a computational perspective, the model was developed in TensorFlow, a popular Python library for deep learning, which scales across a

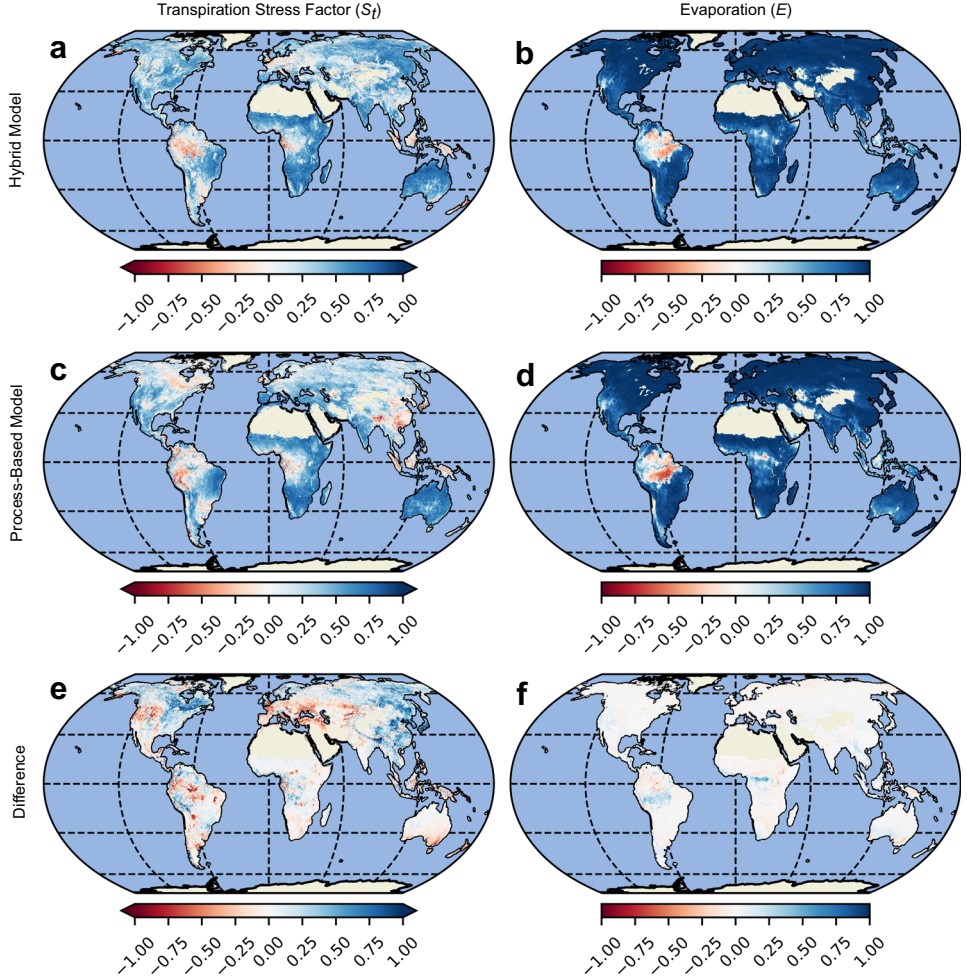

**Fig. 6 Correlation of modeled transpiration stress and evaporation with global datasets. a, c** Comparison of correlation maps for transpiration stress factor ($S_t$) between processed-based and hybrid models with observational *SIF/PAR*. **b** and **d**, Comparison of correlation maps for evaporation ($E$) between processed-based and hybrid models with machine learning-based estimates (FLUXCOM). **e** Difference between **a**, **c**. **f** Difference between **b** and **d**.

wide range of hardware, operating systems, and programming languages. Therefore, the transpiration stress model is agnostic of the host model, and hence can be embedded in different global scale Earth system models.

## Methods

**Evaporative stress formulation in the process-based model.** In the conventional, process-based GLEAM, the total evaporative stress ($S$) is composed of $S_t$ and $S_b$. $S_t$ is defined as

$$S_t = \sqrt{\frac{VOD}{VOD_{max}}\left(1 - \left(\frac{w_c - w_w}{w_c - w_{wp}}\right)^2\right)} \qquad (1)$$

where $VOD_{max}$ is the maximum (99$^{th}$ percentile) VOD, $w_c$ is critical soil moisture, $w_w$ is the soil moisture content of the wettest soil layer, $w_{wp}$ is wilting point. $S_t$ is calculated separately for tall and short vegetation.

$S_b$ is defined as

$$S_b = 1 - \left(\frac{w_c - w_1}{w_c - w_r}\right) \qquad (2)$$

where $w_1$ is the surface soil moisture (first layer in the soil water balance module of GLEAM) and $w_r$ is the residual soil moisture content. The values of $w_{wp}$, $w_c$, and $w_r$ are taken from version 3 of GLEAM[42].

**Development of the deep learning-based transpiration stress formulation.**
The first step consists of defining the target variable, and the appropriate predictors

or covariates. Here, the target variable is the tower-scale $S_t$, calculated as

$$S_t = \frac{E_t}{E_{pt}} \qquad (3)$$

where, $E_t$ is actual transpiration and $E_{pt}$ is potential transpiration.

To estimate $E_t$ in Equation (3), we use daily in situ measurements of $E$, assembled from a total of 557 flux towers. These towers were compiled from FLUXNET[55] (https://fluxnet.org/data/fluxnet2015-dataset/), FLUXNET-CH4 (https://fluxnet.org/data/fluxnet-ch4-community-product/), AmeriFlux (https://ameriflux.lbl.gov/), European Eddy Fluxes Database Cluster (http://www.europe-fluxdata.eu/), and the Integrated Carbon Observation System (ICOS) (https://www.icos-cp.eu/). After the removal of inconsistent values, we end up with 368 stations, out of which 231 stations (approximately 173,000 data points) are classified as having dominantly short vegetation and 137 stations (approximately 103,000 data points) are classified as tall vegetation (refer to Fig. 12 and Table 3 in Supplementary Information for site-specific details and for the mapping of flux tower land cover class to tall and short vegetation). To separate $E_t$ from $E$ at the flux stations, we use empirical functions relating the ratio of $E_t$ to $E$ to the leaf area index (LAI) for different vegetation classes[56] (see Section 2 in Supplementary Information). We remove rainy days from the flux tower datasets to minimize the impact of interception loss on the measurements of $E$ and sensor errors during rain. The LAI-based $E_t$ partitioning model is used here to ensure that the deep learning model of $S_t$ is completely independent from the $E$ partitioning model used to estimate $E_t$ at the eddy covariance sites. Other commonly used partitioning models apply water use efficiency and surface conductance as the main predictors in their empirical approaches[57], which are in turn dependent on vapor pressure deficit (VPD), an important covariate used in the deep learning model developed in this study (see below). Therefore, to prevent such confounding dependencies between VPD and $S_t$, we use an LAI-based empirical model[56]. We note here that none of the existing $E_t$ partitioning models, simple or complex, are perfect. The LAI-based method used here has been validated over different ecosystems[24].

To mitigate the effects of the uncertainty in $E_t$ estimates arising from the choice of the partitioning model used in this study, we supplement the estimates of tall vegetation $E_t$ partitioned from $E$ at the flux towers with a more direct estimate of $E_t$ from sap flow measurements. These in situ measurements are sourced from SAPFLUXNET, a global database of tree-level sap flow measurements[58]. It contains sub-daily time series of sap flow accompanied by in situ-measured hydrometeorological variables and ancillary site, stand and plant metadata. Here, tree-level sap flow data ($cm^3/h$) from SAPFLUXNET version 0.1.5 was expressed per unit projected crown area ($A_c$), estimated as a function of tree trunk basal area at breast height ($A_b$), site mean annual temperature ($MAT$) and precipitation ($MAP$). This model ($\log A_c = -2.53 + 6.02E - 01 * \log A_b + 9.60E - 02 * MAT - 5.48E - 05 * MAP, N = 1055, R^2 = 0.89$) was fitted using data from the Biomass And Allometry Database[59] (BAAD). Tree-level averages of sap flow per unit crown area were then averaged per measured species, weighed by the basal area composition of the stand, and aggregated into daily values. A total of 90 experimental sites are used in the study (Fig. 13 and Table 4 in Supplementary Information). With the addition of the sap flow measurement sites to the 137 flux towers, the total number of data points available for training and validating the deep learning model of $E_t$ stress for tall vegetation is approximately 125,000 (20% corresponding to sap flow sites).

Next, we obtain from daily values of $E_{pt}$ for each station from GLEAM. GLEAM uses a Priestley-Taylor formulation to calculate $E_{pt}$ which has been shown to be generally accurate at ecosystem scales[24]. To account for the scale mismatch between grid-scale estimates of GLEAM and point-scale measurements at the flux tower sites, we scale the $E_{pt}$ values with $E_t$ values using days following rain days as:

$$E_{pt}^{scaled} = \left(\frac{E_{pt}^{raw} - E_{pt,mean}^{raw}}{E_{pt,sd}^{raw}}\right) * E_{t,sd}^{flux} + E_{t,mean}^{flux} \qquad (4)$$

where $E_{pt}^{raw}$ is the raw GLEAM $E_{pt}$ for the specific flux tower site, $E_{pt,mean}^{raw}$ is the mean of the raw GLEAM $E_{pt}$ estimates for the specific flux tower site, $E_{pt,sd}^{raw}$ is the standard deviation of the raw GLEAM $E_{pt}$ for the specific flux tower site, $E_{t,mean}^{flux}$ is the mean of the observed $E_t$ at the flux tower, and $E_{t,mean}^{flux}$ is the standard deviation of the observed $E_t$ at the flux tower. Inherent in this bias-correction approach is the assumption that ecosystems transpire at their potential on days after rainfall.

The covariates used for modeling $S_t$ are the absolute values and seasonal anomalies of the following variables: (a) $PAW$, (b) $VPD$, (c) $T_a$, (d) $SW_i$ (e) $VOD$, (f) $CO_2$. $PAW$ is commonly defined[60] as

$$PAW = \frac{w_w - w_{wp}}{w_c - w_{wp}} \qquad (5)$$

The absolute values and anomalies of $PAW$ for the flux tower sites are derived from GLEAM[42](see section 3 in Supplementary Information for input data used in GLEAM). $VPD$ is derived from relative humidity and $T_a$ data sourced from Atmospheric Infrared Sounder (AIRS) aboard the Aqua satellite mission[61]. $SW_i$ is derived from the Clouds and the Earth's Radiant Energy System (CERES) satellite mission[62]. $VOD$ is derived from the Vegetation Optical Depth Climate Archive (VODCA) dataset[63]. The $SW_i$ and $VOD$ from the same data sources are used as forcing to the GLEAM model to generate $PAW$ to ensure consistency. $CO_2$ data is sourced from the Copernicus Atmopsheric Monitoring Service Global Inversion of Greenhouse Gas Fluxes and Concentrations project (https://ads.atmosphere.copernicus.eu). Finally, within the GLEAM soil water balance model, Equation (5) is solved for short and tall vegetation separately and aggregated based on the fraction of tall and short vegetation in every grid cell. For tall (or short) vegetation flux tower sites, $PAW$ weighted by the corresponding tall (or short) vegetation fraction is extracted. In GLEAM, for tall vegetation, $w_w$ is calculated based on three soil layers, and for short vegetation $w_w$ is based on two soil layers. Here, we note that the choice of estimating the covariates from global gridded datasets rather than in situ measurements at the flux towers and sap flow sites is deliberate. This is done to maintain consistency between the datasets which are used for training (at the point scale) and prediction within the hybrid model (at a coarser scale of 0.25° × 0.25°). In doing so, we aim to minimize the uncertainties that would arise from training and predicting with different datasets. This experiment design choice trades potentially higher local scale prediction and interpretability for more consistent and reliable prediction at the global scale.

**Deep learning model architecture and training**. Designing an optimal deep learning model involves optimizing a number of model-related variables (hyper-parameters) such as the number of layers, number of neurons in each layer, the activation functions in each layer, the rate of dropout to prevent over-fitting, the optimal learning rate, and a loss or objective function along with an appropriate validation metric for evaluating the progress of model training. Here, we design the model architecture, optimize the hyper-parameters, and train the deep learning model using TensorFlow version 2.4[64]. To optimize the hyper-parameters, we employ an automated optimization library available in TensorFlow; specifically, a Bayesian optimization procedure with maximization of the Kling Gupta Efficiency (KGE)[65] as both the training objective and validation metric. In training the objective function is implemented as minimization of $1 - KGE$. KGE is selected as it combines correlation, variability bias, and mean bias into a single metric. KGE is

defined as

$$KGE = 1 - \sqrt{(r-1)^2 + \left(\frac{\sigma_{sim}}{\sigma_{obs}} - 1\right)^2 + \left(\frac{\mu_{sim}}{\mu_{obs}} - 1\right)^2} \qquad (6)$$

where $r$ is linear correlation between simulated and observed values, $\sigma_{sim}$ and $\sigma_{obs}$ are standard deviation of simulations and observations, and $\mu_{sim}$ and $\mu_{obs}$ are mean values of simulations and observations.

First, the Bayesian hyper-parameter optimization was carried out for short vegetation data (231 sites). The most optimal deep learning architecture was found after approximately 1000 iterations of the Bayesian optimization procedure. The resulting deep learning architecture was manually tuned. The final model was then trained for short vegetation $S_t$ with a training:validation data split of 85:15, a batch size of 100, a learning rate of 0.000142, and a maximum epoch size of 1000. The training process does not make any distinction between the different sites—all the 173,000 data points from the 231 sites are treated equally. The training was automatically stopped when the validation objective function started degrading (while the training objective function keeps improving), a sign that the model is overfitting (Fig. 11 in Supplementary Information shows the evolution of the objective during the training process). The same model architecture and training setup was used for training the model for tall vegetation $S_t$ (227 sites). As the model performed satisfactorily with some minor changes, the time consuming hyper-parameter optimization procedure was not performed separately for the tall vegetation dataset (see Fig. 8 in Supplementary Information for the final deep learning models).

**Calculation of SIF/PAR**. SIF data is sourced from the contiguous Orbiting Carbon Observatory-2 (OCO-2) dataset, which is available at 0.05° resolution and 16-day time step[66]. This dataset uses machine learning to gap-fill SIF data to produce a spatially continuous dataset from the OCO-2 satellite, which has a smaller foot-print and infrequent overpass times. The data was spatially aggregated to 0.25° and temporally aggregated to monthly timescales for calculating the correlation maps (Fig. 6) and to seasonal time scales Fig. 4. PAR data is from the CERES mission[62]. PAR data is available at 1.0° resolution at hourly to monthly resolution. Here, the monthly PAR data was used to normalize SIF data.

## Data availability
The outputs of the hybrid model generated in this study and data required for reproducing the results and figures in the main text have been deposited at https://doi.org/10.5281/zenodo.5886608[67].

## Code availability
The deep learning-based formulations of transpiration stress for tall and short vegetation and all the codes required for reproducing the results and figures in this study are available at https://doi.org/10.5281/zenodo.6343005[68].

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

## Acknowledgements

The study was funded by the European Union Horizon 2020 Programme (DOWN2-EARTH, 869550) and the Belgian Science Policy Office (BELSPO) STEREO III program (ALBERI, SR/00/373). DGM acknowledges support from the European Research Council (DRY–2–DRY, 715254). RP acknowledges support from the Spanish State Research Agency (DATAFORUSE, RTI2018-095297-J-I00) and the Alexander von Humboldt Foundation (Germany). This research was possible thanks to the various publicly available flux tower and sap flow measurement networks. The computational resources and services used in this work were provided by the VSC (Flemish Supercomputer Center), funded by the Research Foundation, Flanders (FWO), and the Flemish Government.

## Author contributions

A.K. and D.G.M. conceived and designed the study. A.K. developed the deep learning-based stress formulations, implemented it in GLEAM, and conducted all the model runs. D.R. led the Python implementation of the process-based model (GLEAM). P.H. contributed to the collection and processing of flux tower data. R.P. provided the processed sap flow measurements. A.K. and D.G.M. analyzed the results and wrote the manuscript with inputs from D.R., P.H., and R.P.

## Competing interests

The authors declare no competing interests.
