## [Peer Review File · Nature Communications]

A Deep Learning-Based Hybrid Model of Global Terrestrial EvaporationReviewers' Comments:

Reviewer #1:

Remarks to the Author:

Integrating machine learning approaches and process-based models is a promising solution to more accurate projection of future ecosystem eco-hydrological changes, but remains technically challenging to date. The authors made some pioneering efforts to move this forward. They developed a hybrid framework that can embed machine learning results in the process-based models, and demonstrated its ability to better predict terrestrial evapotranspiration. The framework proposed in this study holds good promise for possible application in next-generation Earth system models. The paper is methodologically sound, well written, and the results are compelling. I raised a few remarks about some aspects of the methodology and the presentation of the results, and my suggestion of publication is expected after some efforts of revision.

Major comments:

The transpiration is obtained from flux tower sites based on an empirical relationship with leaf area index. However, this may not truly capture plant transpiration changes that are controlled primarily by plant physiological traits such as stomatal conductance, particularly under stressed conditions when LAI may not change much.

Further, a set of environmental factors, including VOD, PAW, VPD, T_a , radiation and atmospheric CO₂ concentration places observational constraints on St . My concern is that if St cannot realistically represent responses to stressed conditions, the deep learning model established using the data may overly rely on the seasonal and cross-site relationships between St and environmental factors. Meanwhile, I'm confused why atmospheric CO₂ is considered as a "stress driver" and is included directly in the model. Different to climatic factors, the effect of CO₂ on plant structure and physiology evolves slowly and will not influence inter-annual and shorter-term fluctuations of transpiration. This slowly evolving effect cannot be detected from the flux data with very limited temporal coverage.

The authors compared the performance of transpiration stress predicted by the hybrid model and the original GLEAM model, in terms of the mean bias, variability and correlation with site observations. A key message presently lacking in the manuscript is how incorporating deep learning into process-based model could impact the long-term changes of transpiration (and thus evapotranspiration) over past decades, and its short-term responses to water- or heat-stressed conditions. This is of particular interest to modelers and scientists working on eco-hydrology. If a higher sensitivity is captured by the hybrid model constrained by observations of environmental stressors, we can expect transpiration to change more strongly to future climate with increasing risk of climate extremes.

The normalized SIF by PAR is used to assess the St performance at the global scale (Lines 232-233). I understand that SIF/PAR and the St have similar spatial patterns since they are proportionally related through water-use efficiency. However, under higher CO₂ and extreme conditions, there is much observational evidence of possible decoupling of the carbon and water fluxes with a changing stomatal conductance. Therefore, I'm not sure SIF/PAR is an appropriate variable to validate changes of the transpiration stress.

Minor comments:

Lines 46: "exist" should be "exists"?

Lines 52-58: I would prefer to express this as physically based E_p formulation (Penman or Priestley equation), because E_p is not actual water loss impacted vegetation or atmospheric processes.

Lines 364-395: The following information is important but currently missing from the texts: Why you use VPD, T_a and radiation from satellite observations for the model training, rather than variables

available from flux tower measurements? Did you treat temporal and site samples equally when establishing the deep learning model? What is the temporal resolution of the variables used for model training (I guess this is daily to match GLEAM simulations)?

Reviewer #2:

Remarks to the Author:

In this study, the authors attempted to estimate global terrestrial evaporation using a deep learning-based hybrid model. The hybrid model is made up of two components: a process-based host model and machine learning-based sub-models embedded in the host for representing certain processes. The authors evaluated the performance of the hybrid model and also compared its performance against several flux tower databases and other evaporation products (e.g. FLUXCOM). Comparisons against in situ data and satellite-based proxies demonstrate an enhanced ability to estimate transpiration stress and evaporation globally. The deep learning-based hybrid model is potentially an attractive model to generate a more reliable evaporation. However, this study has some major weaknesses. This manuscript is unsuitable for Nature Communications.

Major comments:

1. Schematic of the hybrid terrestrial evaporation model is questionable. Vegetation transpiration is a complex process and perhaps a deep learning method can estimate more accurate transpiration than process-based model. But, the process of biophysical regulations should not be ignored. If you ignore this process, you can use a pure deep learning method to train evapotranspiration (including vegetation transpiration and soil evaporation). Thus, the accuracy of evapotranspiration estimation may be higher than your hybrid terrestrial evaporation model.
2. Although you compared FLUXCOM gridded product, you only compared the difference between the hybrid and process-based models against in situ data and ignored the difference between the hybrid and a deep learning model against in situ data. In fact, you should illustrate the advantages of your hybrid model versus process-based models and a pure deep learning model at site scale. Otherwise, your conclusions could lead the authors to conclusions that are somehow arbitrary.
3. Although you used a data processing method to account for the scale mismatch between grid-scale estimates of GLEAM and point-scale measurements at the flux tower sites, your method still exists large uncertainties because the effects of landscape heterogeneity on global evapotranspiration is large. Your method may lead to large errors at different scales.
4. The most complex problem is how to acquire the real ET value for a gridded pixel as your training data of machine learning methods. You use the ET values from site to train you model to upscaling global scale may be questionable because the footprint of site vary from several meters to hundreds meters, while global gridded pixel is more than hundreds km. This key problem should be discussed in this manuscript.
5. The training data mainly include FLUXNET databases. However, it is unclear how the training process makes use of each of these datasets. Details should be provided in support materials.
6. There are some results embedded in the Discussion. This discussion should stick with discussion.

Minor comments:

The language should be improved. For example, "algorithms", "formulations", and "model" are need unification. This is just a simple example. The language of the entire manuscript should be carefully checked.

Response to Reviewers

A Deep Learning-Based Hybrid Model of Global Terrestrial Evaporation

Reviewer comments are in **black**

Author responses are in **blue**

Reviewer #1 (Remarks to the Author):

Integrating machine learning approaches and process-based models is a promising solution to more accurate projection of future ecosystem eco-hydrological changes, but remains technically challenging to date. The authors made some pioneering efforts to move this forward. They developed a hybrid framework that can embed machine learning results in the process-based models, and demonstrated its ability to better predict terrestrial evapotranspiration. The framework proposed in this study holds good promise for possible application in next-generation Earth system models. The paper is methodologically sound, well written, and the results are compelling. I raised a few remarks about some aspects of the methodology and the presentation of the results, and my suggestion of publication is expected after some efforts of revision.

Author Response: We thank the reviewer for their insightful comments. We have carefully considered and addressed all the concerns raised by the reviewer.

Major comments:

The transpiration is obtained from flux tower sites based on an empirical relationship with leaf area index. However, this may not truly capture plant transpiration changes that are controlled primarily by plant physiological traits such as stomatal conductance, particularly under stressed conditions when LAI may not change much.

Author Response: We thank the reviewer for this comment. We agree that the LAI-based model used for separating transpiration from evaporation at the flux sites is simple. The choice aimed, however, to avoid more complex methods that use the same variables that are subsequently used as covariates in training the deep learning model. For example, more complex partitioning methods, such those using water use efficiency, are dependent on vapor pressure deficit (VPD), which is also an important covariate in training the deep learning model. This might result in confounding and unrealistic dependencies between VPD and transpiration stress in the deep learning model. In using LAI, we keep the deep learning model as independent as possible from transpiration partitioning.

Nevertheless, all evaporation partitioning algorithms have their own shortcomings, and rely on assumptions regarding the variables used for partitioning and their interactions (Stoy et al. 2019). Given these shortcomings, we incorporated in the revised study novel

transpiration estimates from sap flow measurements. The recent SAPFLUXNET (Poyatos et al. 2021) database collates sap flow measurements from a large network of *in situ* measurement stations across the globe. Sap flow measurements provide a more direct estimation of transpiration than eddy covariance towers, and do not require the use of evaporation partitioning algorithms. Just as the eddy covariance data, sap flow measurements are used to estimate transpiration stress (the target variable for the deep learning model) as well as at the validation stage. The new results show that the deep learning model performs well in representing the transpiration stress at the SAPFLUXNET sites as well. We believe that this adds credence to the ability of the deep learning model in representing the range of conditions in which transpiration is stressed (i.e., below the atmospheric potential), and in addition it makes the results less dependent on the errors caused by the (eddy covariance) evaporation partitioning method.

Manuscript Changes: The rationale for using an LAI-based transpiration partitioning model is now described in the manuscript (Lines 364-373) (reproduced here):

“...The LAI-based E_t partitioning model is used here to ensure that the deep learning model of S_t is completely independent from the E partitioning model used to estimate E_t at the eddy covariance sites. Other commonly used partitioning models apply water use efficiency and surface conductance as the main predictors in their empirical approaches (Stoy et al. 2019), which are in turn dependent on vapor pressure deficit (VPD), an important covariate used in the deep learning model developed in this study (see below). Therefore, to prevent such confounding dependencies between VPD and S_t , we use an LAI-based empirical model (Wei et al. 2017). We note here that none of the existing E_t partitioning models, simple or complex, are perfect. The LAI-based method used here has been validated over different ecosystems (Maes et al. 2019) ...”

Description of the sap flow measurements and how they are integrated into the deep learning model is now included in the manuscript. All the figures (Figures 2-6) have been redone based on simulations from the sap flow-integrated hybrid model. The description of the sap flow measurements from the SAPFLUXNET database (Lines 374-389) is reproduced here:

*“To mitigate the effects of the uncertainty in E_t estimates arising from the choice of the partitioning model used in this study, we supplement the estimates of tall vegetation E_t partitioned from E at the flux towers with a more direct estimate of E_t from sap flow measurements. These *in situ* measurements are sourced from SAPFLUXNET, a global database of tree-level sap flow measurements (Poyatos et al. 2021). It contains sub-daily time series of sap flow accompanied by *in situ*-measured hydrometeorological variables and ancillary site, stand and plant metadata. Here, tree-level sap flow data (cm³/h) from SAPFLUXNET version 0.1.5 was expressed per unit projected crown area (A_c), estimated as a function of tree trunk basal area at breast height (A_b), site mean annual temperature*

(MAT) and precipitation (MAP). This model ($\log(Ac) = -2.53 + 6.02E-01\log(Ab) + 9.60E-02*MAT - 5.48E-05*MAP$, $N = 1055$, $R^2=0.89$) was fitted using data from the Biomass And Allometry Database (BAAD). Tree-level averages of sap flow per unit crown area were then averaged per measured species, weighed by the basal area composition of the stand, and aggregated into daily values. A total of 90 experimental sites are used in the study (Figure 13 and Table 4 in Supplementary Information)."*

Further, a set of environmental factors, including VOD, PAW, VPD, Ta, radiation and atmospheric CO₂ concentration places observational constraints on St. My concern is that if St cannot realistically represent responses to stressed conditions, the deep learning model established using the data may overly rely on the seasonal and cross-site relationships between St and environmental factors.

Author Response: We thank the reviewer for this comment. This comment, combined with the subsequent one, may highlight a small misunderstanding derived from the specific usage of the term "stress" in this study. We emphasize here that "stress" in this study refers to transpiration stress, which is responsible for reducing transpiration from its 'potential' values based on the atmospheric evaporative demand. This use of the terminology is frequently encountered in literature (see e.g., Fisher et al. 2008; Michel et al. 2016). Therefore, 'stress' here does not refer to (or not only to) the sub-optimal physiological state of plants or ecosystems, which we believe is what the reviewer's comment alludes to.

Manuscript Changes: We made numerous changes in the manuscript to clarify the specific usage of the term "stress" in our study. We made sure that we use the term "transpiration stress" rather than just "stress" whenever the term is employed. In addition, we explicitly state that transpiration stress may not be always equated to the physiological stress experienced by vegetation (again, focusing on physiological stress is not the intention of our study) (Lines 69-72):

"...Here, we note that the focus of this study is the stress which limits vegetation transpiration below the atmospheric potential, and therefore can be triggered even under conditions in which plants do not experience stress from a physiological standpoint..."

Meanwhile, I'm confused why atmospheric CO₂ is considered as a "stress driver" and is included directly in the model. Different to climatic factors, the effect of CO₂ on plant structure and physiology evolves slowly and will not influence inter-annual and shorter-term fluctuations of transpiration. This slowly evolving effect cannot be detected from the flux data with very limited temporal coverage.

Author Response: Again, we believe that this concern arises from the different definitions of "stress", as clarified above. In regards to CO₂, we agree that CO₂ concentrations will not

influence the plant structure and physiology at daily timescales. However, here we are concerned with transpiration stress in the sense of divergence from the atmospheric potential based on available energy, which is unequivocally influenced by CO₂ concentrations in as long as they affect stomatal response. In the models of stomatal conductance used in current Earth System Models, atmospheric CO₂ concentration is an important prognostic variable (Leuning, 1995; Medlyn et al. 2011). Further, the slowness of the response of transpiration to CO₂ relates mainly to the CO₂ concentration itself changing slowly. We agree that the slowly evolving effects of CO₂ such as long-term ecological or trait adaptation to CO₂, which we believe is what the reviewer refers to here, may not be adequately captured in the flux tower and sap flow measurements data which has a maximum of two decades of data. However, this should not preclude its inclusion in the model as a deep learning model can be continuously improved as more data are available.

Manuscript Changes: We have added a statement in the manuscript clarifying that the training dataset may not adequately capture the slowly evolving effects of CO₂ on plant adaptation (Lines 160-163):

"...(d) atmospheric carbon dioxide (CO₂) concentration, which exhibits a first order control on stomatal opening. We note that the slowly evolving effects of long-term ecological or trait adaptation to CO₂ may not be adequately captured within the available flux tower and sap flow data..."

The authors compared the performance of transpiration stress predicted by the hybrid model and the original GLEAM model, in terms of the mean bias, variability and correlation with site observations. A key message presently lacking in the manuscript is how incorporating deep learning into a process-based model could impact the long-term changes of transpiration (and thus evapotranspiration) over past decades, and its short-term responses to water- or heat-stressed conditions. This is of particular interest to modelers and scientists working on eco-hydrology. If a higher sensitivity is captured by the hybrid model constrained by observations of environmental stressors, we can expect transpiration to change more strongly to future climate with increasing risk of climate extremes.

Author Response: We agree with the reviewer that analyzing long-term trends in transpiration (and its primary drivers), as simulated by the hybrid model, and quantifying the effect of different stressors at shorter time-scales would be of interest to the ecohydrologic community. This is in fact the subject of an ongoing study in our group. However, presenting such an analysis here would detract from the primary objective of the current work: to present the development of this unique deep learning-based formulation of transpiration stress and its seamless integration with a process-based model at global scales. We believe that the current study presents a novel framework that

can be extrapolated to other models and other domains in climate science, and thus has a strong value on its own. Hence, we believe as well that a comprehensive analysis of trends and disentangling the drivers of transpiration stress at short and long-term time scales deserves a separate study.

Manuscript Changes: No changes have been made in the manuscript in response to this comment.

The normalized SIF by PAR is used to assess the S_t performance at the global scale (Lines 232-233). I understand that SIF/PAR and the S_t have similar spatial patterns since they are proportionally related through water-use efficiency. However, under higher CO₂ and extreme conditions, there is much observational evidence of possible decoupling of the carbon and water fluxes with a changing stomatal conductance. Therefore, I'm not sure SIF/PAR is an appropriate variable to validate changes of the transpiration stress.

Author Response: We agree with the reviewer that SIF/PAR may not be a perfect diagnostic across the entire range of transpiration stress in response to external factors. However, currently SIF/PAR is arguably the best observational proxy of S_t that is spatio-temporally continuous and available globally (Pagán et al. 2019). We think such a comparison is relevant at the monthly time scales at which it has been done in this study.

Manuscript Changes: We have added a sentence of caution stating that the comparison between SIF/PAR and transpiration stress may not be appropriate under extreme environmental conditions and higher CO₂ (see Lines 264-266):

"... We also caution that the comparison may not be appropriate under extreme conditions and higher CO₂, where carbon and water cycles may decouple..."

Minor comments:

Lines 46: "exist" should be "exists"?

Author Response: Thank you. This sentence has been reframed.

Manuscript Changes: The line (Lines 54-56) now reads as:

"... Independent of the approach, significant uncertainties remain in the current global estimates of E..."

Lines 52-58: I would prefer to express this as physically based E_p formulation (Penman or Priestley equation), because E_p is not actual water loss impacted vegetation or atmospheric processes.

Author Response: We now refer to E_p formulations as ‘process-based’ rather than ‘least empirical part’. We are of the opinion that even Penman is not completely physically-based as the surface and aerodynamic resistances cannot be estimated without some level of empiricism. Therefore “process-based” seems more appropriate.

Manuscript Changes: The line (Lines 61-65) now reads as:

“...Nevertheless, the chosen E_p function forms the most process-based part of the stress-based E models...”

Lines 364-395: The following information is important but currently missing from the texts: Why you use VPD, T_a and radiation from satellite observations for the model training, rather than variables available from flux tower measurements?

Author Response: This is a very good point that should have been clarified; indeed, the *in situ* data could be used for this purpose. The primary reason for using satellite-based observations of the covariates, rather than flux tower measurements, is to maintain consistency with the scale at which the deep learning model is applied within the hybrid model afterwards ($0.25^\circ \times 0.25^\circ$). In doing this (using gridded satellite-based data for covariates instead of local scale measurements for training), we minimize the potential errors which would arise from using gridded inputs for prediction (within the hybrid model). We note here that this design choice trades local scale prediction power (not the intention of the study) for more consistent and reliable prediction at the coarse gridded scale.

Manuscript Changes: We made this clear in the manuscript (Lines 424-432):

*“...Here, we note that the choice of estimating the covariates from global gridded datasets rather than *in situ* measurements at the flux towers and sap flow sites is deliberate. This is done to maintain consistency between the datasets which are used for training (at the point scale) and prediction within the hybrid model (at a coarser scale of $0.25^\circ \times 0.25^\circ$). In doing so, we aim to minimize the uncertainties that would arise from training and predicting with different datasets. This experiment design choice trades potentially higher local scale prediction and interpretability for more consistent and reliable prediction at the global scale...”*

Did you treat temporal and site samples equally when establishing the deep learning model? What is the temporal resolution of the variables used for model training (I guess this is daily to match GLEAM simulations)?

Author Response: Yes, we make no distinction between the different sites when establishing the model (the only distinction is between tall and short vegetation). The temporal scales of the deep learning model and GLEAM are the same (daily).

Manuscript Changes: The time scale of model training has now been clearly stated in the manuscript (Lines 149-151):

"...Deep learning models are developed for tall and short vegetation separately at daily time scales..."

We also make clear that the training process does not make any distinction between the sites (Lines 456-457):

"...The training process does not make any distinction between the different sites - all the 173,000 data points from the 231 sites are treated equally..."

Reviewer #2 (Remarks to the Author):

In this study, the authors attempted to estimate global terrestrial evaporation using a deep learning-based hybrid model. The hybrid model is made up of two components: a process-based host model and machine learning-based sub-models embedded in the host for representing certain processes. The authors evaluated the performance of the hybrid model and also compared its performance against several flux tower databases and other evaporation products (e.g. FLUXCOM). Comparisons against in situ data and satellite-based proxies demonstrate an enhanced ability to estimate transpiration stress and evaporation globally. The deep learning-based hybrid model is potentially an attractive model to generate a more reliable evaporation. However, this study has some major weaknesses. This manuscript is unsuitable for Nature Communications.

Author Response: We thank the reviewer for their insightful comments. We have carefully considered and addressed all the concerns raised by the reviewer and we hope that by doing so the manuscript is now deemed suitable for publication in the journal.

Major comments:

1. Schematic of the hybrid terrestrial evaporation model is questionable. Vegetation transpiration is a complex process and perhaps a deep learning method can estimate more accurate transpiration than a process-based model. But, the process of biophysical regulations should not be ignored.

Author Response: By biophysical regulations, we believe that the reviewer refers to the influence of plant traits such as root depth, isohydricity, species-specific biogeochemical strategies, leaf area index, and other anatomical and morphological traits. In such a case,

we agree with the reviewer that including biophysical information would improve the deep learning formulation of transpiration stress. However, incorporating such information into a global scale model is hindered by the lack of global dynamic observations. Notwithstanding this, we are of the firm opinion that the deep learning-based formulation of transpiration stress presented here represents a substantial improvement over existing state-of-art stress formulations which generally use very few stress drivers (for example, only soil moisture), which are often assumed to be linearly inter-related.

Manuscript Changes: We now clearly state that the impact of plant traits described above has not been incorporated in this version of transpiration stress (Lines 165-169):

"...In addition, the influence of plant traits such as root depth, isohydricity, and other anatomical and morphological traits, and their fine-scale or inter-species variations is not explicitly considered, since reliable data for upscaling such traits so that they can be implemented within a global model is not available..."

If you ignore this process, you can use a pure deep learning method to train evapotranspiration (including vegetation transpiration and soil evaporation). Thus, the accuracy of evapotranspiration estimation may be higher than your hybrid terrestrial evaporation model.

Author Response: As the reviewer has raised important questions regarding the difference between hybrid models and purely deep learning models, we wish to clarify that the primary objective of developing a hybrid model is distinctly different from that of a purely deep learning model (which is used primarily for prediction). Our study shows that deep learning can be used to mimic the transpiration stress response of different ecosystems, and has the potential to replace existing formulations which have been developed with local-scale (often laboratory-based) experiments. We believe that such formulations can be implemented in other process-based models, including land surface models and Earth system models (such as Noah-MP for example, which uses a stress-based formulation for estimating actual transpiration). Additionally, within a hybrid model the deep learning model is physically constrained by the water and energy balance limitations imposed by the process-based part of the model, an important distinction which separates them from purely machine learning models. The latter are unconstrained by any physical limits and thus are only useful as a predictive tool. Hybrid models are, hence, potentially more reliable for examining sensitivities of transpiration stress to external factors.

Manuscript Changes: We believe that the introduction in the manuscript makes a convincing case for hybrid models, clearly stating the advantages of such models over purely machine learning-based models.

2. Although you compared FLUXCOM gridded product, you only compared the difference between the hybrid and process-based models against in situ data and ignored the difference between the hybrid and a deep learning model against in situ data. In fact, you should illustrate the advantages of your hybrid model versus process-based models and a pure deep learning model at site scale. Otherwise, your conclusions could lead the authors to conclusions that are somehow arbitrary.

Author Response: We thank the reviewer for this useful comment. We believe that a comparison to the FLUXCOM gridded product (purely machine learning-based model) at the flux towers would indeed be a good addition.

Manuscript Changes: We now present the comparison between the performance of the hybrid model and FLUXCOM at the flux towers and sap flow sites in the Supplementary Information (figure 2 for violin plots, figure 9 for site-wise comparison). While the results are presented in the supplementary section, they are also summarized in the main text:

Description of the violin plots (Lines 216-220): *"...We also compare the estimates of E from the hybrid model with those of a purely machine learning-based dataset, FLUXCOM (Figure 2 in Supplementary Information). We see that while the overall performance of both approaches is similar, the hybrid model tends to outperform FLUXCOM in forest (tall vegetation) regions..."*

Description of the sitewise comparison (Lines 245-249): *"...Finally, we also compare the performance of the hybrid model against FLUXCOM at individual flux tower and sap flow measurement sites (Figure 9 in Supplementary Information). Similar to the comparison with the process-based model, we see that the hybrid model underperforms in the relatively arid western parts of the US and Iberia..."*

3. Although you used a data processing method to account for the scale mismatch between grid-scale estimates of GLEAM and point-scale measurements at the flux tower sites, your method still exists large uncertainties because the effects of landscape heterogeneity on global evapotranspiration is large. Your method may lead to large errors at different scales.

Author Response: We agree that the mismatch between the grid-scale of GLEAM and the small footprint of flux tower sites will lead to uncertainties. However, we emphasize here the fact that we are not training the deep learning directly on observed evaporation at the flux towers, unlike pure machine learning-based approaches like FLUXCOM. Here our main objective is to develop a better formulation of transpiration stress and therefore, as long as the *in situ* observations are an accurate representation of the vegetation stress response of the vegetation (either tall or short), the scale of measurements will not have a large adverse effect. The hybrid model then relies on the process-based component to

resolve some of the sub-pixel spatial heterogeneity using fractional vegetation cover and the soil water balance model.

Manuscript Changes: We now explicitly state that the target of the deep learning model within the hybrid is different from that of pure machine learning-based approaches such as FLUXCOM: here deep learning is used to develop a universal transpiration stress formulation using flux tower data (Lines 112-119):

"...We exploit recent progress in satellite-based remote sensing and an unprecedented number of in situ observations spread across the globe to develop a novel formulation of S_t from the ground-up without any prior assumptions. Therefore, the objective of using deep learning is fundamentally different (development of an improved formulation of the transpiration stress response of vegetation) from that of purely machine learning-based models which are designed to predict E_t directly and suffer from issues of scale..."

For explanation about how hybrid models account for sub-grid heterogeneity better than purely machine learning-based models, please refer to the response to the next comment.

4. The most complex problem is how to acquire the real ET value for a gridded pixel as your training data of machine learning methods. You use the ET values from site to train you model to upscaling global scale may be questionable because the footprint of site vary from several meters to hundreds meters, while global gridded pixel is more than hundreds km. This key problem should be discussed in this manuscript.

Author Response: Again, we believe that this is actually one of the biggest advantages of the hybrid approach. As opposed to using the deep learning model for directly modeling transpiration at the flux tower scale (which would suffer issues of scale), our objective is to derive an accurate transpiration stress formulation which can then be used by a process-based model to derive evaporation at the grid scale. These formulations can be derived at the local vegetation scale (as is routinely done in developing existing transpiration stress formulations). Here, we develop the S_t formulations at the footprint of the FLUXNET towers. Therefore, as long as the observed evaporation (transpiration) is an accurate representation of the response of vegetation to external stressors, the issues of scale do not affect our application as much as it would a purely machine learning-based approach. In addition, GLEAM models transpiration and bare soil evaporation separately and combines them per pixel. Therefore the learnt transpiration stress function only affects the transpiration estimates of GLEAM. The bare soil evaporation within the pixel is calculated separately from transpiration, which would not have been possible if the training was done on grid-scale evaporation.

Manuscript Changes: In the revised manuscript, the objective of using deep learning in this study (as opposed to purely machine learning-based studies) is described better (Lines 112-119):

“...We exploit recent progress in satellite-based remote sensing and an unprecedented number of in situ observations spread across the globe to develop a novel formulation of S_t from the ground-up without any prior assumptions. Therefore, the objective of using deep learning is fundamentally different (development of an improved formulation of the transpiration stress response of vegetation) from that of purely machine learning-based models which are designed to predict E_t directly and suffer from issues of scale...”

We clearly state that the difference in the footprint between the in situ observations and the model is a drawback in purely machine learning-based models (Lines 94-97):

“...More importantly, the use of pure machine learning methods for specifically estimating E_t at global scales is hindered by the fact that in situ observations of E_t have a small footprint which is not representative of E_t at the coarser grid scales at which global models operate...”

We have also added a new schematic in Figure 1 (reproduced below) which shows how the hybrid model takes into account sub-grid heterogeneity by estimating transpiration (using the deep learning-based stress formulation for short and tall vegetation) and bare soil evaporation (using the process-based stress formulation) separately, and then aggregating the total evaporation per grid cell. This is also described in the text (Lines 132-137):

“...GLEAM simulates E as a summation of its constituents: E_t , bare-soil evaporation (E_b), and interception loss (E_i). E_t and E are estimated for every grid cell of the global model using a Priestley Taylor-based formulation for E_p and their respective evaporative stress factors (S_t and S_b), weighted by the fractional coverage of short vegetation, tall vegetation, bare-soil, and open water (Figure 1)...”

5. The training data mainly include FLUXNET databases. However, it is unclear how the training process makes use of each of these datasets. Details should be provided in support materials.

Author Response:

The details of the training data, including the sources of the target (S_t) and the covariates, and the training process (for example, hyperparameter choices, training-validation splits) are described in the Methods section within two subsections: *Development of the deep learning-based transpiration stress formulation* and *Deep learning model architecture and training*. We gently request the reviewer to clarify the specific additional details that are required.

Manuscript Changes: In response to the comments of the reviewer, and reviewer #1, we have also incorporated transpiration estimates from a global database of sap flow measurements (SAPFLUXNET, Poyatos et al. 2021). Sap flow measurements provide a more direct estimate of transpiration than eddy-covariance towers, and the validation results show that the deep learning model adequately represents transpiration stress as observed in the sap flow measurements. The description of the sap flow measurements from the SAPFLUXNET database (Lines 374-389) is reproduced here:

“To mitigate the effects of the uncertainty in E_t estimates arising from the choice of the partitioning model used in this study, we supplement the estimates of tall vegetation E_t partitioned from E at the flux towers with a more direct estimate of E_t from sap flow measurements. These in situ measurements are sourced from SAPFLUXNET, a global database of tree-level sap flow measurements (Poyatos et al. 2021). It contains sub-daily time series of sap flow accompanied by in situ-measured hydrometeorological variables and ancillary site, stand and plant metadata. Here, tree-level sap flow data (cm³/h) from

SAPFLUXNET version 0.1.5 was expressed per unit projected crown area (A_c), estimated as a function of tree trunk basal area at breast height (A_b), site mean annual temperature (MAT) and precipitation (MAP). This model ($\log(A_c) = -2.53 + 6.02E-01\log(A_b) + 9.60E-02*MAT - 5.48E-05*MAP$, $N = 1055$, $R^2=0.89$) was fitted using data from the Biomass And Allometry Database (BAAD). Tree-level averages of sap flow per unit crown area were then averaged per measured species, weighed by the basal area composition of the stand, and aggregated into daily values. A total of 90 experimental sites are used in the study (Figure 13 and Table 4 in Supplementary Information)."*

6. There are some results embedded in the Discussion. This discussion should stick with discussion.

Author Response: We thank the reviewer for this comment; we agree this partitioning could be improved.

Manuscript Changes: In the revised manuscript, we have made sure that the discussion is completely separate from the results. The following sentences were deleted: *"...Specifically, the biggest improvements in St are seen in northern latitudes, likely due to the consideration of incoming radiation (a key driver of stomatal conductance). On the contrary, the deep learning-based St tends to overestimate the stress in tropical rainforests, primarily in the DJF season..."* and *"...Further, the estimates of E from the hybrid model accurately capture the temporal dynamics and the spatial patterns of E seen in both the in situ network of flux tower observations and a (purely) machine learning-based dataset (FLUXCOM)..."*.

In their place, the following sentence was added (Lines 318-321):

"...The biggest improvement is seen in forested (tall vegetation) regions, especially in the northern latitudes. This has important implications for constraining transpiration estimates in tropical, temperate, and boreal forests which contribute a major part of the global transpiration..."

Minor comments:

The language should be improved. For example, "algorithms", "formulations", and "model" are need unification. This is just a simple example. The language of the entire manuscript should be carefully checked.

Author Response: We thank the reviewer for pointing this out.

Manuscript Changes: We have unified these terminologies in the revised manuscript where applicable. We now use the term "algorithm" to refer only to satellite retrieval algorithms as it is a standard terminology in the remote sensing community. We use the

term “model” to refer to the GLEAM model which consists of different sub-modules. To clearly distinguish the sub-modules from the host “model”, we use the term “formulation” to refer to the different equations used for calculating the stress (empirical stress formulation in the process-based model and the machine learning-based stress formulation in the hybrid) and potential evaporation (Priestley-Taylor formulation). We have also edited the text and improved the language throughout the manuscript.

References:

Fisher, J. B., Tu, K. P., & Baldocchi, D. D. (2008). Global estimates of the land-atmosphere water flux based on monthly AVHRR and ISLSCP-II data, validated at 16 FLUXNET sites. *Remote Sensing of Environment*, *112*(3), 901-919.

Leuning, R. (1995). A critical appraisal of a combined stomatal-photosynthesis model for C3 plants. *Plant, Cell & Environment*, *18*(4), 339-355.

Medlyn, B. E., Duursma, R. A., Eamus, D., Ellsworth, D. S., Prentice, I. C., Barton, C. V., ... & Wingate, L. (2011). Reconciling the optimal and empirical approaches to modelling stomatal conductance. *Global Change Biology*, *17*(6), 2134-2144.

Michel, D., Jiménez, C., Miralles, D. G., Jung, M., Hirschi, M., Ershadi, A., Martens, B., McCabe, M. F., Fisher, J. B., Mu, Q., Seneviratne, S. I., Wood, E. F., and Fernández-Prieto, D.: The WACMOS-ET project – Part 1: Tower-scale evaluation of four remote-sensing-based evapotranspiration algorithms, *Hydrol. Earth Syst. Sci.*, *20*, 803–822, <https://doi.org/10.5194/hess-20-803-2016>, 2016.

Pagán, B. R., Maes, W. H., Gentine, P., Martens, B., & Miralles, D. G. (2019). Exploring the potential of satellite solar-induced fluorescence to constrain global transpiration estimates. *Remote Sensing*, *11*(4), 413.

Poyatos, R., Granda, V., Flo, V., Adams, M. A., Adorján, B., Aguadé, D., ... & van der Tol, C. (2021). Global transpiration data from sap flow measurements: the SAPFLUXNET database. *Earth System Science Data*, *13*(6), 2607-2649.

Stoy, P. C., El-Madany, T. S., Fisher, J. B., Gentine, P., Gerken, T., Good, S. P., ... & Wolf, S. (2019). Reviews and syntheses: Turning the challenges of partitioning ecosystem evaporation and transpiration into opportunities. *Biogeosciences*, *16*(19), 3747-3775.

Reviewers' Comments:

Reviewer #1:

Remarks to the Author:

I find the authors have addressed sufficiently most of my comments. I only have two remaining minor comments that require further revision (as below). From my perspective, I recommend the publication of this MS at Nature Communications.

I can't say that I agree "a deep learning model can be continuously improved if more data are available". I'm not convinced that the effects of rising CO₂ can be (fully) captured by the deep learning model. For example, it is well known that other global ET products upscaled from eddy covariance sites with machine learning (e.g., doi: 10.1038/nature09396) often fail to capture the effect of rising water use efficiency under rising CO₂, no matter whether CO₂ is included as a predictor or not. If the authors can test whether the inclusion of CO₂ will make a difference, that would be fine.

Response to Reviewers

A Deep Learning-Based Hybrid Model of Global Terrestrial Evaporation

Reviewer comments are in **black**

Author responses are in **blue**

Reviewer #1 (Remarks to the Author):

I find the authors have addressed sufficiently most of my comments. I only have two remaining minor comments that require further revision (as below). From my perspective, I recommend the publication of this MS at Nature Communications.

Author Response: We thank the reviewer for the positive and constructive comments over the two rounds of peer review.

I can't say that I agree "a deep learning model can be continuously improved if more data are available".

Author Response: We agree with the reviewer that availability of more data does not guarantee a better deep learning model. Being accurate, it should have read "deep learning predictions tend to improve as more data are available".

Manuscript Changes: No manuscript changes, as this sentence was only included in the response to reviewers document during the previous round of review.

I'm not convinced that the effects of rising CO₂ can be (fully) captured by the deep learning model. For example, it is well known that other global ET products upscaled from eddy covariance sites with machine learning (e.g., doi: 10.1038/nature09396) often fail to capture the effect of rising water use efficiency under rising CO₂, no matter whether CO₂ is included as a predictor or not. If the authors can test whether the inclusion of CO₂ will make a difference, that would be fine.

Author Response: We largely agree with the reviewer. The effect of long-term rising [CO₂], and the associated low-frequency changes in sensitivity of transpiration stress, may not be fully captured by the current version of the deep learning model, which only uses a couple of decades of *in situ* data for training. However, [CO₂] also varies at shorter time scales, and has an ample seasonal cycle. That, together with the fact that [CO₂] concentration is an important input in different stomatal models (Leuning, 1995; Medlyn et al. 2011), we believe that the use of [CO₂] in an expert-guided machine learning model of transpiration stress (as the one developed here) is justified.

Based on the suggestion of the reviewer, we present here the validation statistics (Kling-Gupta Efficiency) of the deep learning models for short and tall vegetation when [CO₂] is removed from the set of input data. By comparing the statistics to those from the default model (which uses CO₂ as an input) we show that the consideration of [CO₂] has a mild but significant positive effect on the performance of the deep learning models, for both short and tall vegetation (5.7% improvement in KGE for short vegetation and 5.9% for tall vegetation). This shows that the deep learning models are capable of capturing some of the influence of [CO₂] on transpiration, but this does not necessarily mean that the long-term effects on WUE the reviewer alludes to are captured.

	KGE for Short Vegetation	KGE for Tall Vegetation
Hybrid Model With CO ₂	0.776	0.662
Hybrid Model Without CO ₂	0.734	0.625

Note: Higher KGE implies a better model

Manuscript Changes: We included a cautionary statement which makes it clear that the deep learning model developed here may not be able to capture the long-term effects of [CO₂] such as the effect of rising [CO₂] on water use efficiency (Lines 161–165).

“...We note that the slowly evolving effects on transpiration of long-term ecological or plant trait adaptation in response to rising CO₂ (as reflected on water use efficiency trends) may not be adequately captured by training the machine learning algorithms on the limited record length of available flux tower and sap flow measurements (Jung et al. 2010)...”

References

Jung, M. et al. Recent decline in the global land evapotranspiration trend due to limited moisture supply. *Nature* 467, 951–954 (2010)

Leuning, R. (1995). A critical appraisal of a combined stomatal-photosynthesis model for C3 plants. *Plant, Cell & Environment*, 18(4), 339–355.

Medlyn, B. E., Duursma, R. A., Eamus, D., Ellsworth, D. S., Prentice, I. C., Barton, C. V., ... & Wingate, L. (2011). Reconciling the optimal and empirical approaches to modelling stomatal conductance. *Global Change Biology*, 17(6), 2134–2144.